# CD36-Fatty Acid-Mediated Metastasis via the Bidirectional Interactions of Cancer Cells and Macrophages

**DOI:** 10.3390/cells11223556

**Published:** 2022-11-10

**Authors:** Noorzaileen Eileena Zaidi, Nur Aima Hafiza Shazali, Thean-Chor Leow, Mohd Azuraidi Osman, Kamariah Ibrahim, Wan-Hee Cheng, Kok-Song Lai, Nik Mohd Afizan Nik Abd Rahman

**Affiliations:** 1Department of Cell and Molecular Biology, Faculty of Biotechnology and Biomolecular Sciences, Universiti Putra Malaysia, Serdang 43400, Malaysia; 2Institute of Tropical Forestry and Forest Products, Universiti Putra Malaysia, Serdang 43400, Malaysia; 3Department of Biomedical Science, Faculty of Medicine, University of Malaya, Kuala Lumpur 50603, Malaysia; 4Faculty Health and Life Sciences, INTI International University, Persiaran Perdana BBN, Putra Nilai, Nilai 71800, Malaysia; 5Health Sciences Division, Abu Dhabi Women’s College, Higher Colleges of Technology, Abu Dhabi 41012, United Arab Emirates

**Keywords:** CD36, metabolism, macrophage, tumour microenvironment, metastasis

## Abstract

Tumour heterogeneity refers to the complexity of cell subpopulations coexisting within the tumour microenvironment (TME), such as proliferating tumour cells, tumour stromal cells and infiltrating immune cells. The bidirectional interactions between cancer and the surrounding microenvironment mark the tumour survival and promotion functions, which allow the cancer cells to become invasive and initiate the metastatic cascade. Importantly, these interactions have been closely associated with metabolic reprogramming, which can modulate the differentiation and functions of immune cells and thus initiate the antitumour response. The purpose of this report is to review the CD36 receptor, a prominent cell receptor in metabolic activity specifically in fatty acid (FA) uptake, for the metabolic symbiosis of cancer–macrophage. In this review, we provide an update on metabolic communication between tumour cells and macrophages, as well as how the immunometabolism indirectly orchestrates the tumour metastasis.

## 1. Introduction

Cancer cells exploit profound remodelling in their metabolism, with vast tumour cells favouring their metabolism towards aerobic glycolysis, a phenomenon exemplified by the Warburg effect [1,2]. Cancer cell metabolism thus serves as a broader paradigm for the hallmarks of cancer. In contrast to normal cells, cancer cells increase their metabolic activity to tailor their high energy demands, accelerated proliferation, and adaptation to evolving cellular roles and biological functions [3,4]. Thus, an enhanced level of tumour metabolic environment may be exploited or hindered by a vast number of mechanisms intrinsically linked to cancer progression including colonizing new metastatic niches and triggering protumour immune responses.

Under dynamic metabolism, these metabolic fluxes simultaneously influence both the composition of the local inflammatory milieu and the function of tumour-infiltrating immune cells. In these settings, a tumour can exploit the metabolic cancer–immune cell crosstalk to facilitate the development of a tumour-permissive environment via the secretion of metabolic by-products or oncometabolites. Despite diverse signalling cues within the tumour microenvironment (TME), among which cancer metabolites are available, tumour-associated macrophages (TAMs) are generally associated with cancer-mediated metabolism as cancers have developed a mechanism to favour the activation of protumour macrophages that contribute to pro-tumorigenic processes and immunosuppressive responses.

One of the prominent features related to structural insights into metabolic activity is cellular receptors, which act as master regulators integrating signalling molecules to initiate internal signalling pathways. Cluster of differentiation 36 (CD36) among which is a widely expressed transmembrane glycoprotein is widely expressed and is involved in lipid metabolism and signalling. However, this CD36 receptor has been detected in both immune cells such as macrophages and neutrophils and cancer cells [5,6]. The purpose of this review is to discuss the contribution roles of the CD36 receptor as a bridge for cancer–macrophage metabolic symbiosis. Here, we focus on how the metabolic shift in cancer and its metabolites shape functional macrophage phenotypes, and conversely, how the immunometabolism of macrophages orchestrates the complex process of tumorigenesis and antitumour immune response. We also discuss CD36-mediated metabolisms in these key cells.

## 2. CD36 Receptor: Gene, Structure, Distribution and Function

### 2.1. CD36 Discovery and Structure-Function

In the early 1970s, the discovery of CD36 began when Kobylka and Carraway identified a membrane protein in breast epithelial cells that could not be proteolyzed in milk fat globules [7]. Later, in 1987, that molecule was described as platelet glycoprotein IV (GP IIIb or GP IV) [8] and reported to be a protein that mediates thrombospondin-1 (TSP-1) binding in platelets [9]. Subsequently, CD36 was implicated as a macrophage receptor for the oxidation of low-density lipoprotein (ox-LDL) [10]. Since then, CD36 has been widely established as a scavenger receptor and is putatively involved in membrane protein facilitating fatty acid (FA) transport. The extensive studies into the role of CD36 in fatty acid and lipid metabolism in health and disease were initiated by Abumrad, Grimaldi and colleagues [11].

CD36, also referred to as FA translocase (FAT), is a heavily N-linked glycosylated 80-kDa integral protein membrane encoded by the CD36 gene that is located on chromosome 7q21.11 and covers 72 kilobases, including 19 known exons [12]. CD36 belongs to the class B scavenger receptor family (SCARB3) and lysosomal integral membrane protein II (LIMP-II) [13,14]. The protein CD36 is often described as having a ‘hairpin-like’ configuration (Figure 1). Structurally, CD36 harbours two transmembrane segments leading to two short cytoplasmic tails (N- and C- terminal) that are separated by a large glycosylated extracellular loop containing a hydrophobic sequence for ligand-binding sites [15]. Furthermore, two palmitoylation sites on both N- and C- hydrophobic stretches regulate CD36 localization to lipid raft membranes to optimize the cellular fatty acid uptake. The extracellular loop containing disulphide bonds between cysteine residues are essential for intracellular processing, maturation and transport [16,17]. The large hydrophobic cavity serves as a channel through which ligands are transported from the extracellular space to the outer leaflet of the phospholipid bilayer of membrane. Neculai and colleagues’ crystallization studies provided functional insights into fatty acid transport by CD36, whereby the extracellular domains of CD36 form a hydrophobic cavity into which the fatty acids can be translocated to and from the membrane bilayer [18].

CD36 is a multiligand receptor because of its high affinity for many ligands. The interaction between distinct ligands and the CD36 surface receptor in specific cell types was shown to mediate signalling pathways. Ligand of CD36 can be classified as lipid-related ligands such as long-chain FAs (LCFAs), ox-LDL and oxidized phospholipids (ox-PLs) or as protein-related ligands including TSP-1 and TSP-2. As FA translocase, CD36 plays a role in the uptake and membrane transport of LCFAs. CD36 participates in atherosclerotic lesion formation due to its substantial capacity to bind and endocytose ox-LDL into macrophages and impairs adipocyte insulin in a CD36-dependent manner [19,20]. While CD36 has often been implicated in adhesion and scavenging functions, data presented by Dawson and colleagues showed that antiangiogenic TSP-1 associated with CD36 induces the inhibition of both migration and tube formation by TSP-1 [21]. Moreover, CD36 can also associate with other transmembrane proteins such as integrins, an interaction believed to be crucial for apoptotic cell engulfment. Further studies reported that macrophages phagocytose apoptotic cells via an integrin system including phosphatidylserine (PS) and α_v_β_3_, which on human macrophages is associated with CD36 [22]. In innate immunity, CD36 also acts as a pattern-recognition receptor (PRR) by recognizing molecular patterns that are associated with pathogens [23]. It also helps in the clearance of cell debris and phagocytosis [24].

### 2.2. CD36 Distribution and Functions

As a surface protein, CD36 is highly distributed in several cell types and tissues, such as in adipocytes, macrophages, mitochondria, adipose tissue, skeletal muscles and cardiac muscles. Distribution of CD36 has been extensively linked to delivery of FAs in adipocytes via CD36-mediated endocytosis [25]. These studies proved that cell surface CD36 is exclusively localized within lipid rafts. Therefore, the disruption of lipid rafts is necessary to inhibit LCFAs in adipocytes. Excess FAs taken up by CD36 are primarily converted to triacylglycerol (TAG) storage as cytoplasmic lipid droplets in adipose tissues while in skeletal muscles they are utilized in fatty acid oxidation (FAO) [26,27]. Macrophage ingests ox-LDL by binding and internalization in antigen-presenting cells [28], whereas dendritic cells phagocytose apoptotic cells both via the CD36 receptor [29,30]. Interestingly, CD36 also has been found to express on human skeletal muscle mitochondria membrane, where it is responsible for mitochondrial LCFAs transport and subsequent oxidation [31]. Enhanced CD36 expression and consequent elevated LCFAs uptake and TAG accumulation can contribute to lipid accumulation in cardiac muscle cells and skeletal muscle [32], insulin resistance and type 2 diabetes [33].

It is possible that the CD36 receptor is abundant in cancer cells lines, tumour tissues and their adjacent normal tissues including stromal cells and immune cells. CD36 is increasingly emerging as a favourable prognostic biomarker of tumour, primarily in epithelial origin tumours such as acute myeloid leukaemia, breast cancer, cervical cancer, colorectal cancer, gastric cancer, glioblastoma, hepatocellular carcinoma, oral squamous carcinoma, ovarian cancer, pancreatic cancer and prostate cancer (Table 1). Not unexpectedly, most tumours shift toward de novo FA synthesis. They can also exploit multiple ways to scavenge lipids and extensively rewire their metabolism. This mechanism is to sustain their membrane biosynthesis during rapid proliferation, to balance energy storage and energy expenditure and generate signalling molecules.

## 3. Metabolic Reprogramming and Metastasis

Cancer metastasis is an intricate process and a complex phenomenon that has been extensively studied, yet metabolic reprogramming controlling this highly inefficient metastatic cascade has not been widely explored. The invasion–metastatic cascade is the consequence of cancer cells’ undergoing metabolic alterations and adaptations to thrive nutrient starvation and interference with metabolic and immune profiles in circulation and metastasis TMEs [51]. In these settings, the level of tumour metabolic environment may be exploited or hindered by a vast number of mechanisms intrinsically linked to cancer progression, including colonization of new metastatic niches and immunosurveillance escape. Metabolic reprogramming could promote metastasis by (1) the upregulation of oncometabolites and enzymes involved in metastasis-signalling cascades; (2) adipocyte–cancer crosstalk leads to metabolic demands, thus allowing cancer cells prone to invasion and metastasis; and (3) metastasis is facilitated by metastasis-associated macrophages via secreted cytokines and oncometabolites.

### 3.1. CD36 Responds to Exogenous Fatty Acids

Generally, fatty acids (FAs) are biomolecules that are involved in an abundance of cellular events [52], such as providing substrates for energy production [53], developing cell membrane structure and modulating signalling pathways [54]. Highly proliferative cancer cells require FAs to support cell growth [55,56], disseminate [57,58,59], regulate membrane assembly [60], activate proliferative signalling and meet bioenergetic requirements [60,61]. In mammalian cells, FAs can be acquired through direct exogenous uptake from the local surrounding niche, or via activation of the de novo synthesis pathway using nutrients such as glutamine and glucose. During metastatic progression onset, the rapid-growing cancer cells often leads to the need for an ever-increasing blood supply, resulting in hypoxia and nutrient deprivation. Studies have shown that cancer cells acquire the increased uptake of exogenous FAs to compensate for reduced glucose de novo FA synthesis and to sustain their FA metabolism during conditions of metabolic stress.

For instance, Pascual et al. reported that PA enhances the metastatic potential in CD36^+^ metastasis-initiating oral squamous cell carcinomas, in a CD36-dependent manner. In addition, CD36-dependent lymph node metastases increased in size and frequency upon being treated with PA, without affecting the primary tumour growth [45]. The finding is consistent with Pan et al., whose experiments established CD36 as a key mediator of FA-mediated metastasis in gastric cancer. They demonstrated that CD36 promotes migration and invasion in gastric cancer via uptake of exogenous PA and activation of GSK-3β/β-catenin signalling [42]. However, Jiang et al. demonstrated that PA promoted metastasis in gastric cancer and induced CD36 expression through activating the hexosamine biosynthetic pathway (HBP). They also observed that O-GlcNAcylation promotes CD36 transcription by activating the NF-κB pathway and consequentially enhances FA uptake [62]. Importantly, recent research indicates that oral carcinomas and melanomas in mice fed a palm-oil-rich diet and tumour cells that were exposed briefly to PA in vitro remained highly metastatic even after serial transplantation into secondary recipient mice [63]. Specifically, the depletion or knockdown of CD36 in oral squamous carcinoma cells led to a loss of PA-induced prometastatic memory. Notably, increased intratumoural Schwann cells population activated by metastatic cells downstream of dietary PA is correlated with the CD36^+^ metastatic signature. Tao et al. reported additional evidence to verify the regulatory role of CD36 on hepatocellular carcinoma in vivo. Their results reveal that CD36 accelerates proliferation and metastasis of hepatocellular carcinoma by enhancing FA absorption through AKR1C2 and jointly affect the FA metabolism.

The relevance of exogenous FA and CD36-mediated metastasis was further supported by findings from Zhang et al., laboratory [64]. They described that oleic acid (OA) promotes metastasis and proliferation of colon carcinoma cells HCT116. Further experiments proved that dietary OA upregulated CD36 expression which promotes tumour growth and metastasis in cervical cancer cells HeLa [38]. Their results indicated that OA promotes metastasis and growth of cervical cancer by inducing CD36-dependent activation of Src kinase and downstream of the ERK1/2 pathway. Collectively, these studies suggest that cancer cells can compensate for impaired FA synthesis by enhancing exogenous FA uptake by upregulating fatty acid transporter CD36 to perform invasion and metastasis. In the case of studying colorectal cancer, overexpression of CD36 is associated with a significant increase in invasion and metastasis [39]. They also observed that CD36 regulates MMP28 expression, which is inversely associated with E-cadherin expression. Based on these findings, it seems likely that CD36 promotes colorectal cancer metastasis by upregulating MMP28 and E-cadherin cleavages.

In the context of angiogenesis, the correlation of CD36 and VEGF receptor 2 with angiogenic switch is observed markedly reduced in the absence of TSP-1. Interestingly, TSP-1 acquires sequential activation of CD36 to attenuate angiogenesis and therefore inducing apoptosis or blocking VEFGR2 pathway in the endothelial cells [65]

### 3.2. CD36 and Metabolic Symbiosis

Reprogramming the metabolism of fatty acids is increasingly being recognized as essential for cancer cells within tumour compartments to comply with metabolic symbiosis. It is well established that the immediate metabolic environment within a tumour is constantly modified by the symbiotic relationships between cancer cells and stromal cells. For instance, the metastatic dissemination of breast cancer is correlated with the secretion of breast-associated adipocytes via induced CD36 expression. Hereby, adipocytes profoundly influence the invasiveness and metastasis of malignant breast cancer by inducing metabolic reprogramming by elevating the expression of CD36, accompanied by accelerated FA uptake. Intriguingly, STAT3 and ERK1/2 are activated by an upregulated expression of CD36 receptors which is required for adipocyte-induced epithelial–mesenchymal transition (EMT) [66]. Thus, the activation of STAT3 signalling may account for the enhancement of metastasis through the upregulation of MMP9 and TWIST [67]. Wang et al., reported that breast cancer tissue located adjacent to adipose tissue expressed a high level of CD36 and fatty acid transport protein-1 (FATP1). Upon adipocytes co-cultivated with breast cancer cells, the expression of CD36 and FATP1 was elevated compared to adipocytes cultivated alone.

Adipocyte-ovarian cancer co-culture was demonstrated to express high levels of CD36, which coincided with an increased cellular FA uptake and lipid accumulation in ovarian cancer [47]. Importantly, Ladanyi et al., reported that the inhibition of CD36 expression in ovarian cancer resulted in a significantly reduced of both baseline and adipocyte-stimulated invasion and migration in a xenograft mouse model. However, Mukherjee et al., findings described the significant role of FABP4 in regulating adipocyte-mediated metastasis in ovarian cancer [68]. Their findings showed ovarian cancer cells cocultured with adipocytes had elevated CD36 and FABP4 expression, whereas knockout of FABP4 resulted in inhibition of several tumorigenic pathways including proliferative and migration in ovarian cancer cells. Further evidence establishing the link between adipocytes and cancer, CD36^+^ oral squamous cell carcinoma stimulated by high-fed diet or adipocyte-cultured medium strongly elevated pro-metastatic potential [45]. In accordance with previous studies, the regulatory role of phosphatidylinositol transfer protein, cytoplasmic 1 (PITPNC1) in adipocytes and gastric cancer omental metastasis was discovered. Tan et al. found that PITPNC1-mediated FA metabolic reprogramming was regulated by co-cultured omental adipocytes and consequently facilitated gastric cancer omental metastasis in CD36-dependent manner [69].

In recent years, pre-clinical and clinical evidence highlights the significant role of lipid metabolism as a major influence on both immune and clinical responses of cancer patients. Current cancer nano-immunotherapy protocols are based on the precision therapy and cancer diagnose therapy. For instance, hyaluronic acid nanoparticles or iron-oxide nanomedicines could be used as diagnostic or therapeutic tool in cancer [70]. Additionally, nano-immunotherapy aims to reprogram both specific and innate immune responses, to enhance the intrinsic anti-tumoural activity. Pharmacological reprogramming of lipid metabolism in tumour-associated macrophages showed efficacy in suppressing tumour immunosuppression and cancer immunotherapies. Therefore, strategies to dampen TAMs’ immunosuppressive M2-like signature are of high interest to boost the efficacy in cancer immunotherapy research. One promising strategy is to target CD36 as therapeutic potential of metabolic interventions on the complex modulation of lipid metabolism in both CD36-faciliated cancer metastasis and TAMs immunosuppression.

## 4. Involvement of CD36 and Macrophages in Metastasis

In 1863, Rudolph Virchow proposed that the inflammatory profiles present on-site of the tumour lesions in the inflamed tissue area which are abundantly infiltrated by immune cells may contribute to cancer development [71]. His findings noted the correlation between chronic inflammation and the development of cancer by discovering the presence of leukocytes in neoplastic tissues. Over the past decades of cancer research, a comprehensive understanding of inflammation and TME has increasingly been investigated. The persistence of chronic inflammation is one of the important characteristics of tumour progression that promotes all stages of tumorigenesis from malignant to the establishment of tumour invasion and metastasis. In fact, the TME is highly infiltrated by a broad spectrum of immune cells which acquire specialized functions and phenotypes. Tumour-associated immune cells comprise T-cells, dendritic cells, B-cells, macrophages, neutrophils, and natural killer cells [72]. Among of these cells, tumour-associated macrophages (TAMs) stand out. An approximate 50% of the macrophage’s population has been detected in solid tumours and been confirmed to have a fundamental pro-tumoural role.

Generally, macrophages phenotypic heterogeneity and plasticity are reflected in their gene expression pattern, specialized tissue-specific functions, and cytokine production, which are orchestrated depending on the activation stimulus. Macrophages can be schematically identified as classically activated, M1 (pro-inflammatory and anti-tumoural phenotype) and alternatively activated, M2 (anti-inflammatory and pro-tumoural phenotype). Macrophages adopt the M1 phenotype following activation by lipopolysaccharides (LPS) and interferon-gamma (IFNγ), and by producing pro-inflammatory cytokines such as tumour necrosis factor alpha (TNF tumour necrosis factor alpha (TNFα), interleukin (IL)-6, IL-12, and IL-1β, nitric oxide (NO) production and exhibiting phagocytic traits. On the contrary, IL-13 and IL-4 have been demonstrated as the stimuli that favour M2 subpopulation polarization, delivering anti-inflammatory cytokines such as transforming growth factor β (TGF-β), IL-10, IL-4, IL-13, IL-8, IL-1Ra, and vascular endothelial growth factor (VEGF), and associated with wound healing and tissue repairing. Importantly, activation of macrophages is a complex and significantly controlled process, which includes the variance of intracellular signalling and transcription pathways. During persistent inflammation within the TME, TAMs in established tumours are usually biased toward the M2-like phenotype-acquired macrophages, which contributes to the initiation of tumour invasion, migration, angiogenesis, and immunosuppression.

TAMs frequently co-exist in the same microenvironment as tumour cells and stromal cells; thus polarization of TAMs is significantly regulated by a heterogenous milieu of microenvironmental cues such as cytokines, chemokines, and other growth factors [73]. It has been established that TAMs resembling M2 phenotypes, secrete immunosuppressive cytokines such as IL-10, prostaglandins, and reactive oxygen species (ROS), VEGF, and induces EMT for invasion and metastasis (Figure 2). These reciprocal interactions between TAMs and metastasis are the consequences of adaption to cellular metabolism by both tumour cells and stromal cells to overcome environmental stress in the circulation and metastatic niche. In fact, TAMs undergo metabolic reprogramming in response to altered tumour cell-derived metabolic cues, through the activation of glycolysis, FA synthesis and amino acid metabolism. Here, we mainly focus on how fatty acid metabolism of TAMs shapes their functional phenotype through CD36 FA receptors.

### 4.1. CD36 Regulated TAMs-Facilitated Metastasis in TME

#### 4.1.1. TAMs and Their Pro-Tumorigenic Functions

As mentioned above, the interaction of tumour cells and adjacent immune cells especially TAMs can greatly impact the tumorigenesis from different aspects. Successful metastasis of cancer cells also hinges on the mechanism of TAMs coupled with their FA metabolism alterations. Macrophages within the TME contribute to a growth-suppressive state, but these cells may later be reprogrammed by the tumour microenvironmental cues within the TME to develop pro-tumorigenic functions. It is noteworthy that FAs are critical metabolites during macrophage polarization. Proinflammatory M1 macrophages synthesize FAs to exploit them as precursors of inflammatory mediators by relying on high glycolytic metabolism to promote rapid ATP generation. On the contrary, anti-inflammatory M2 macrophage displays intact TCA cycle and metabolic modification to enhanced FAO and mitochondrial OXPHOS, which is driven by FA uptake [74].

FA uptake in M2 macrophage occurs via the lipolysis of circulating lipoproteins through CD36 mediation. The internalized FAs can transcriptionally activate the nuclear receptor peroxisome proliferator-activated receptor gamma (PPARγ) and the peroxisome proliferator-activated receptor gamma coactivator-1 beta (PGC-1β) [75]. Mechanistically, the PPARγ activation in M2 macrophages regulates the activation of the oxidative program in these cells. PPARγ acts as a FA sensor and FAO enzyme transcriptional activators. In view of the necessity of FA generation to fuel FAO and the importance of FAO for M2 activation. Huang et al., found that the uptake of TAG substrate via CD36 and its subsequent lipolysis by lysosomal acid lipase (LAL) is more highly expressed in M2 than in M1 macrophages and is vital for full macrophage M2 activation upon IL-4 induction [76]. These findings suggested that the uptake and lipolysis of exogenous TAGs may serve to generate FAO during M2 activation. Likewise, TAMs, which share phenotypic and anti-inflammatory properties with M2 macrophage, were also observed to have high level of FAO. According to Su et al., TAMs accumulate FAs by increasing FA uptakes through CD36 receptor [77]. Their findings implicate that FA accumulation and FAO are both essential for differentiation and pro-tumour functions of TAMs. Interestingly, a recent study found that the pro/anti-tumour functions of TAMs were linked and dependent on the type of FA enriched within the cells. According to Odegaard et al., oleic acid (OA) enhances expression of the anti-inflammatory alternative state (M2) macrophage by activating the nuclear receptor PPARδ [78]. In addition, tumour-promoting inflammation mediated by the upregulation and activation of NLR family pyrin domain containing 3 (NLRP3) inflammasome in macrophages resulted in colorectal cancer [79] and gastric cancer migration [80]. Their findings further supported by Hofbauer et al. studies which demonstrated that activation of NLRP3 inflammasome in TAMs mediated by beta-2-microglobulin (β2m) promotes multiple myeloma progression [81].

#### 4.1.2. CD36-Mediated Lipid Droplet Accumulation in TAMs

Lipid droplets (LDs) are constituted as cellular organelles specialized in neutral lipid storage for metabolic energy and hydrolysis. LDs hubs as energy storage for triacylglycerols (TAGs) released as FAs for mitochondrial oxidation (FAO), making LDs significantly correlated with FA metabolism. In addition, LDs are linked to multiple cellular functions, such as lipid synthesis, protein storage, membrane synthesis and viral replication [82]. As observed in IFN- γ activated macrophages, increased exogenous FAs uptake and inhibition of mitochondrial respiration by iNOS-derived NO are required to induce LDs accumulation. Thus, their study established a novel metabolic pathway whereby carbon atoms in acyl chains of TAG derived from exogenous FAs and glucose provided carbon to the glycerol headgroup of TAG. Other data also indicated long-chain FA metabolism and the accumulation of LDs are strongly linked to the regulatory phenotype of macrophages. The formation of LDs and its derived FAs, specifically unsaturated FAs, facilitates the mitochondrial respiration, hence contributing to the polarization of TAMs [83]. Accordingly, their findings emphasized that oleate and LD-derived FAs facilitate mitochondrial respiration regulates the suppressive phenotype of myeloid cells, in resulting in endogenous FAs contributing to the polarization of CD206^+^ TAMs.

Studies suggest besides regulating macrophage polarization, LDs formation also determines the magnitude of inflammatory response in macrophages attributed to upregulation of hypoxia-inducible LD-associated (HILPDA) protein [84]. Based on van Dierendonck et al.’s findings, the tight regulation of lipid efflux from LDs by the upregulation of HILPDA and the downregulation of adipose triglyceride lipase (ATGL) protein levels, leads to reduced pro-inflammatory precursors and suppresses the production of prostaglandin-E2 (PGE2) and the proinflammatory cytokine IL-6. Despite this, the changes in cellular LD dynamics can result in an advanced macrophage functional activity along as well as a synergistic effect on the differentiation of THP-1 cell line macrophages [85]. The contribution of LDs content by further addition of an external source of FFAs (NA-oleate) to PMA stimulated THP-1 macrophages showed a marked increase in CD68 expression along with higher phagocytosis, ROS and NO generation, and release of proinflammatory cytokines. The stimulation of PMA stimulated THP-1 with an external source of FFAs triggers the release of calcium from the intracellular pool that affects the LD biosynthesis and/or maturation through Rab5a via AKT phosphorylation. Overall, the advantageous consumption of endogenous FAs mediated by CD36 in macrophages directly contributed to the function and polarization of TAMs, and was further mediated by the accumulation of LDs.

#### 4.1.3. TAMs-Mediated Migration and Invasion

During metastasis onset, the recruitment and accumulation of TAMs to the hypoxia metastatic niche will trigger TAM-derived inflammatory cytokines, which in turn are involved in the regulation of the EMT process. For instance, TAMs prime Gas6/Axl-NF-κB activation in both stromal and cancer cells co-culture, which then increase the tumour growth and metastasis in oral cancer cells [86]. Meanwhile, a recent study found EMT high tumours showed significant enrichment of TAMs and their overexpression of immunosuppressive cytokines IL-10 and TGF-β [87]. TME-produced immunosuppressive cytokines have been identified as active precursors supporting inflammatory response and mediating tumour progression and EMT. Biologically, macrophage secretes soluble factors involved in the EMT process such as IL-1β, IL-8, tumour necrosis factor-α (TNF-α) and TGF-β via activation of acyl-CoA synthetase that catalyses the thioesterification of FAs. Apart from cytokines, TAMs also regulated EMT by secreting matrix metalloproteinases (MMPs) including MMP-1, MMP-2, MMP-3, MMP-7 and MMP-9, cysteine cathepsin and serine proteases. These proteolytic enzymes of TAMs are important for hydrolysing the extracellular matrix (ECM), activating growth factors, and promoting angiogenesis. Meanwhile, neovascularization is crucial for supplying essential nutrients and oxygen to the growing tumour, and consequently promotes metastasis. Studies have demonstrated TAMs contribute tumour neovascularization by upregulating angiogenesis-related growth factors by inducing pro-inflammatory mediators such as VEGF, TGF-β, IL-1 and IL-6 [88].

More recently, accumulating evidence suggested that reprogramming TAMs can be achieved by exposing macrophages to endocrine disruptors to induce anti-tumorigenic activity and promote M1 polarization. Lu et al. demonstrated macrophage polarization toward inflammatory M1 phenotype and high secretion of pro-inflammatory cytokines can be stimulated by the exposure bisphenol A (BPA) [89]. Notably, endocrine disruptors such as BPA and phthalate are known as pro-oncogenic drivers in cancer. Quagliariello et al. demonstrated that exposure to BPA co-incubated with doxorubicin increased inflammation in cardio myoblasts [90]. Another study further explored that at very low doses of BPA can induces prostate cancer cells migration [91], and similar effects were observed in breast cancer cells [92,93]. However, the link between BPA and CD36 expression has been determined in non-alcoholic fatty liver disease (NAFLD), which BPA increases hepatic lipid uptake by stimulating ROS-induced CD36 overexpression [94]. Based on this evidence, further studies must be investigated to understand the underlying mechanism and association of BPA with CD36 in cancer progression and immune response.

### 4.2. CD36 Targeted Nano-Immunotherapy

Current approach in the field of cancer nano-immunotherapy have provided new attractive strategies to hamper TAM-driven pro-tumorigenic processes. Rational designs of drug-nanoparticle are specifically developed to effectively controls the ability of TAMs to regulate TME, for the purpose of specific macrophage-targeting against cancer. Kuninty and colleagues present novel “tail-flipping” nanoliposomes to target specifically to M2-like macrophages via bio-mimicking anionic and oxidized phospholipids that are internalized and recognized by macrophages as a natural clearance mechanism [86]. Their studies proved that their nanoliposomes delivered to M2-TAMs via CD36 receptor able to alter their functionality by inhibiting STAT-6 pathway which led to inhibition of macrophage-induced tumour cell migration. Furthermore, the direct effect of these altered TAMs was examined on orthotopic 4T1 breast tumour model. Their findings showed that approximately 70% of tumour growth as well as inhibition of pre-metastatic niche formation in lungs was reduced by muramyl tripeptide (MTP)-manipulated TAMs. In summary, the current nanoliposome systems represent as an effective approach to target and manipulate M2-TAMs, which can be exploited for developing cancer nano-immunotherapy treatment.

In addition, the efficacy of the new era nano-immunotherapy depends on its potential to manipulate the cancer–macrophage interaction and immune factors in TME, including cytokines and inflammatory pathways. As mentioned previously, cytokines are significant to mediate the cancer–macrophage interaction via inflammatory responses, which can either promote or inhibit the growth and metastasis of cancer cells. Therefore, these inflammation-based immunotherapies have been largely supplanted in nanotherapeutics to boost immunostimulatory cytokines. Recently, Gao and colleagues developed a nano-delivery system derived from macrophage membrane coated ROS-responsive nanoparticles (MM-NPs) efficiently bind with CD36 membrane antigens [95]. Macrophage-coated NPs sequester multiple proinflammatory cytokines, suppressing inflammation. In summary, CD36-targeted nano-immunotherapy plays a pivotal role to revolutionize the treatment of cancer.

## 5. Concluding Remarks

We present here the recent findings on the essential role of CD36 receptor not only in controlling cellular FA uptake and utilization and its influence on FA metabolism in cancer cells but also influences the crosstalk with macrophages in TME. Many significant studies had revealed the effects of FAs on the bidirectional communication between tumour cells and immune cells during cancer progression, particularly in metastasis. The evidence for a key metabolic role of CD36 in TAM-mediated metastasis is strong, therefore targeting the related metabolites and FAs involved in the metabolism may provide an emerging approach for cancer treatment and manipulation of TAMs. As mentioned above, TAMs and its alteration in FA metabolism represent a novel approach that may alter the landscape of future immunotherapy in treating cancer.

## Figures and Tables

**Figure 1 cells-11-03556-f001:**
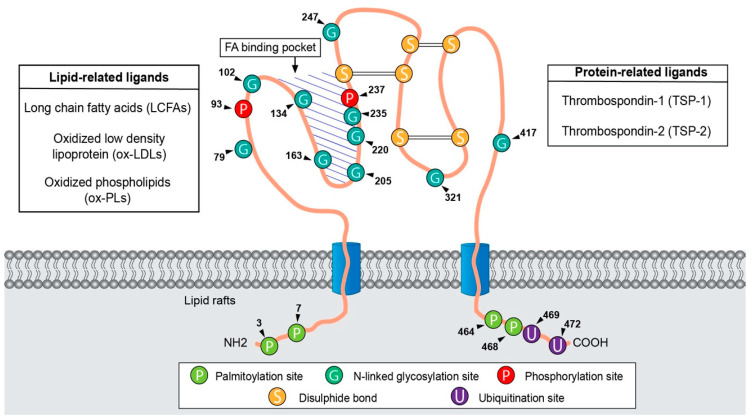
Schematic presentation of the CD36 structure. CD36 harbours two transmembrane domains in a large extracellular region containing ligand-binding sites. CD36 has two cytoplasmic tails (N-terminal and C-terminal) that contains four palmitoylation sites. The C-terminus contains two ubiquitination sites. The large extracellular loop contains 10 N-linked glycosylation sites and two phosphorylation sites. CD36 also contains three disulphide bonds between extracellular cysteines. In addition, the hydrophobic pocket is involved in ligand binding and serves as a tunnel through which hydrophobic ligands are transported from the extracellular space across the phospholipid membrane bilayer. Arrowhead and numbers denote the approximate position of amino acid residues.

**Figure 2 cells-11-03556-f002:**
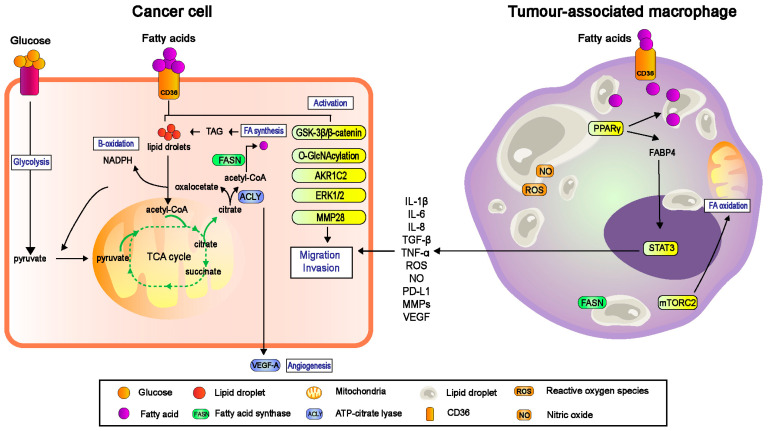
The bidirectional interaction between cancer cells and macrophages regulated by CD36 receptor in promoting metastasis. Due to hypoxic conditions, cancer cells in tumour microenvironment (TME) increase the uptake of exogenous fatty acids for survival, energy source and rapid proliferation. Depending on the type of cancer, literatures have mentioned CD36 promotes metastasis, invasion and angiogenesis by activating the downstream GSK-3β/β-catenin, O-GlcNAcylation, AKR1C2, ERK1/2 and MMP28 signalling pathways. However, the recruitment of macrophage to TME and become tumour-associated macrophages (TAMs) is significant for cancer progression. TAM tends to have a higher level of FA uptake and accumulation via CD36, which accordingly enhance the fatty acid oxidation (FAO) of TAMs to generate more energy. This phenomenon also upregulates the lipid biosynthesis to produce more nitric oxide (NO) and reactive oxygen species (ROS) and secrete high level of immunosuppressive cytokines such as interleukin-6 (IL-6), interleukin-8 (IL-8), tumour necrosis factor-α (TNF-α) and more. The dynamic interaction between cancer cells and TAMs constantly favouring the tumour evasion and ultimately metastasis. NADPH, nicotinamide adenine dinucleotide phosphate; TAG, triacylglycerol; VEGF, vascular endothelial growth factor; MMPs, matrix metalloproteinase; PPARγ, peroxisome proliferator- activated receptor gamma; FABP4, fatty acid-binding protein; STAT3, signal transducer and activator of transcription 3; mTORC2, mammalian target of rapamycin complex 2.

**Table 1 cells-11-03556-t001:** Contributions of CD36 expression in various type of cancer tissues.

Type of Cancer	Contribution of CD36	References
Acute myeloid leukaemia	Increases leukaemia burden and shorten survival in vivo	[34]
Breast cancer	Essential survival mechanism in HER2-positive breast cancer Activates expression of pro-proliferation and migration genes while inhibiting expression of apoptotic genes	[35]
		[36]
Cervical cancer	Promotes the epithelial–mesenchymal transition and metastasis in cervical cancer by interacting with TGF-β Promotes cervical cancer cell growth and metastasis via up-regulating the Src/ERK pathway	[37]
		[38]
Colorectal cancer	Promotes metastasis by increasing MMP28 and decreasing e-cadherin expression Increases in cellular proliferation via upregulation of survivin in CRC cells	[39]
		[40]
Gastric cancer	Promotes peritoneal metastasis via fatty acid uptake	[41]
	Promotes metastasis of gastric cancer via AKT/GSK-3β/β-catenin pathway	[42]
Glioblastoma	Increases glioblastoma progression and tumour initiation in cancer-stem cells	[43]
Hepatocellular carcinoma	Promotes epithelial–mesenchymal transition, enhances migration and invasion	[44]
Oral squamous carcinoma	Initiates and promotes metastasis and worsens prognosis Promotes lymph node metastasis	[45]
		[46]
Ovarian cancer	Omental adipocytes reprogram tumour metabolism due to high exogenous fatty acid uptake	[47]
	Facilitates the proliferation and migration and lymph node metastasis	[48]
Pancreatic cancer	Mediates pancreatic cancer development and progression	[49]
Prostate cancer	Increases cancer cell proliferation and migration, and increase tumour burden in vivo	[50]

## Data Availability

Not applicable.

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
