# Peer review of "CD36-Fatty Acid-Mediated Metastasis via the Bidirectional Interactions of Cancer Cells and Macrophages"

_cells, 2022, doi:10.3390/cells11223556_

Round 1

Reviewer 1 Report

Manuscript titled " CD36-Fatty Acid Mediated Metastasis via Bidirectional Interactions of Cancer Cells and Macrophage" is a very interesting review in the field of oncolgy and fatty acid metabolism. The review is well written, overall structure is of good quality. However, authors should improve the manuscript in some parts:

1. In introduction, authors should explain also how endocrine disruptors like bisphenol A should reduce immune functions against cancer cells and how bisphenol A should modulate fatty acid metabolism in cancer cells as well as in cardiac cells, incresing cardiovascular diseases and cancer cell growth ( cite 10.1016/j.etap.2019.03.006 ).

2. In discussion, authors should describe how nanotechnology should modulate fatty acid metabolism and how liposomes are able to increase immune functions against cancer cells; i.e iron-oxide nanoparticles ( cite 10.1016/j.nano.2016.08.022 )

Author Response

Please see the attachment for the responses to your comments and suggestions. Please use the "Track Changes" feature in the Word document to view the revisions made to the manuscript.

Reviewer 2 Report

The Review “CD36-Fatty Acid Mediated Metastasis via Bidirectional Interactions of Cancer Cells and Macrophages” by Zaidi et al.

The Manuscript by Zaidi et al. reviews the role of CD36 receptor-mediated fatty acid uptake by tumor-associated macrophages on tumor progression and metastasis. The Manuscript is well written and logically organized. The main text includes essential information describing the impact of TAMs on the tumor microenvironment, signal pathways induced by FA metabolites, and other molecular mechanisms of cancer progression. Meanwhile, the figures, especially Fig.2, are too general and lack essential information. Taking into account the general goal of any review paper – to facilitate understanding of a large amount of experimental data, I suggest authors consider the update of existing illustrations or draw an additional figure or table summarizing information from the main text. In conclusion, the authors propose CD36 FA metabolites as a promising target for cancer therapy. This statement is true, but the Manuscript lack information on existing therapeutic approaches or discussion of perspective targets and drugs. I suggest removing the unsupported speculation or adding the missing information for at least the most remarkable points.

Author Response

(The authors gave the same response as above.)

Reviewer 3 Report

The authors present the comprehensive roles of CD36 in macrophages and cancer cells, and the involvement of CD36 and macrophages in metastasis. I have some suggestions to improve the review article. 

Specific comments:

1. Authors described that "CD36 is increasingly emerging as a favourable prognostic biomarker of tumour, primarily in epithelial origin tumours such as ovarian cancer, prostate cancer, breast cancer, colon cancer, oral squamous cell carcinoma, hepatocellular carcinoma, acute myeloid leukaemia, and glioblastoma (lines 137-140)." I suggest that authors specify the references in each tissue.

2. Authors have described "CD36-Mediated Lipid Droplet Accumulation in TAMs" in part 4.1, and given some examples as references 68-71 in this part. Although these papers have shown the essential roles of LDs in activation and inflammatory response of macrophages, it is not clear whether LDs in macrophages are mediated by CD36. Especially, the data from reference 68 showed that anti-CD36 neutralizing antibody did not diminish the increased lipid uptake in macrophages induced by IFNγ. This suggests that IFNγ-induced LDs accumulation in macrophages is not mediated by CD36. Therefore, at least, authors should delete the sentences related with the reference 68 (lines 335-342). I suggest authors could describe the roles of LDs accumulation in TAMs for metastasis in this part.

3. It has been reported that CD36 has anti-angiogenic roles in endothelial cells as described in lines 109-112. However, CD36 seems to induce angiogenesis in Figure 2 and authors have not described the pro-angiogenic roles of CD36. If CD36 works as pro-angiogenic receptor in cancer cells and/or TAMs, authors should define the reference. Moreover, authors should more discuss the relationships between CD36-mediated fatty acid uptake in cancer cells and TAMs and angiogenesis, because angiogenesis is essential event in metastasis.

Minor point:

There are same references (25, 26). Authors should delete either of the reference.

Author Response

(The authors gave the same response as above.)

Round 2

Reviewer 1 Report

Manuscript titled " CD36-Fatty Acid Mediated Metastasis via Bidirectional Interactions of Cancer Cells and Macrophage" is a very interesting original article describing the key role of CD36 pathways in cancer cell growth and metastasis. The manuscript have an overall structure of good quality, figures are good and conclusion are in agree with original hypothesis. However, authors should improve the manuscript in some parts:

1. First, authors should explain the role of endocrine disruptors in macrophage-cancer cell interaction. Considering that bisphenol A and phtalates could increase inflammation in tumour microenvironment, authors should explain the key roles of bisphenol A as pro-oncogenic driver in cancer. Cite 10.1016/j.etap.2019.03.006. 

2. Second, authors should add a paragraph on how nanomedicine could improve cancer microenvironment by modulating cytokines and pro-inflammatory pathways.  For example, how hyaluronic acid nanoparticles or iron-oxide nanomedicines could be used as diagnostic or therapeutic tool in cancer ( cite 10.1016/j.nano.2016.08.022) 

3. Finally, authors should explain the involvement of NLRP3 in cancer microenvironment. 

The manuscript will be acceptable after minor revision. 

Author Response

(The authors gave the same response as above.)

Reviewer 2 Report

The authors have adequately addressed my comments

Author Response

Thank you for your comments. Please use the "Track Changes" feature in the Word document to view the revisions made to the manuscript.

Reviewer 3 Report

Overall manuscript quality has been significantly improved after additional descriptions. However, I cannot understand the response of authors to suggestion point 2. Although authors have argued that previous report supports their discussion, the report has shown the pivotal roles of CD36 in "cardiomyocytes". It is not the roles of CD36 to uptake fatty acids in "TAMs". At least, Rosas-ballina et al., result did not support the involvement of CD36 in increased lipid uptake in "macrophages". If authors remain the description, they should delete the term "CD36-mediated" in subtitle of part4.1. I would support publication if the paper was rewritten to address my concerns.

Author Response

(The authors gave the same response as above.)

Round 3

Reviewer 1 Report

Authors repied in a satisfactory manner. Manuscript is acceptable in the current version

Reviewer 3 Report

The manuscript could be acceptable now.